# Female Sexual Function in Twin Pregnancy

**DOI:** 10.3390/ijerph19063546

**Published:** 2022-03-16

**Authors:** Anna Fuchs, Agnieszka Dulska, Jakub Bodziony, Mateusz Szul, Agnieszka Drosdzol-Cop

**Affiliations:** Chair and Department of Gynecology, Obstetrics and Oncological Gynecology, Medical University of Silesia, 40-055 Katowice, Poland; dulska.agnieszkaz@gmail.com (A.D.); jt.bodziony@gmail.com (J.B.); kezerin11@gmail.com (M.S.); cor111@poczta.onet.pl (A.D.-C.)

**Keywords:** twin pregnancy, sexuality, sexual dysfunction, desire, sexual activity in pregnancy, assisted reproductive technology

## Abstract

Introduction: The incidence of twin pregnancy is estimated at 1 per 80 single pregnancies. As the topic of sexual function among women with multiple pregnancy is insufficiently developed, we believe it is appropriate to raise this subject. Methods: A prospective study was conducted on 100 women during subsequent trimesters of pregnancy. Results: From a group of 100 women, 54 women were primiparous, while 46 women had a history of previous delivery. The mean overall FSFI (female sexual function index) was found to be 24.3 ± 6.1. Mean FSFI was the highest in the first trimester at 25.6. The result decreased successively to 24.8 ± 7 and 22.6 ± 8.4 in the second and third trimesters, respectively. The patient’s place of residence had a crucial impact on their FSFI score. The results were considerably higher for residents of small and medium towns or cities—24.4 ± 3.8 and 25.9 ± 4.9, respectively—while for those living in rural areas, the FSFI reached only 21.7 ± 5.4. Conclusions: The present study shows that the FSFI decreased throughout twin pregnancy. The lowest observed FSFI occurred in the third trimester, while the highest FSFI occurred during the first trimester.

## 1. Introduction

Sexuality used to be associated only with sexual intercourse, an interpretation which does not fulfil its true implication. It is not only a broadly understood physical act in itself, but is also recognized as engendering a general sense of comfort, well-being, safety, and empathy in terms of sexual functioning [1,2].

Unfortunately, being pregnant is still considered as a reductive factor of sexual activity. The main reason for this is due to a lack of sexual desire during pregnancy, which is often accompanied by misinformation and concerns about the well-being of both the fetus and the mother. In this situation, future mothers fear the occurrence of hemorrhage, infectious diseases, premature labor induction, or potential harm to the fetus itself [3]. Even more contradictory information is connected with the topic of sexual activity among women with multiple pregnancies.

It has been confirmed that the incidence of twin pregnancies has doubled over the course of the last fifty years, meaning that twins comprise almost 3% of all births in countries such as France or the USA [4,5]. According to a report in the Demographic Yearbook of Poland 2019, 2.5% of all births during the previous year originated from twin pregnancy [6,7,8].

The fact that almost 20% of multiple pregnancies are connected with a long-term commitment to conceiving a child should not be seen as insignificant [9]. Sexual intercourse sometimes becomes something objectified, mandatory, and often accompanied by a sense of duty, or even worse, necessity. The above-mentioned factors dramatically alter the sexuality of future parents. These couples often seek professional help in order to achieve a long-awaited pregnancy in the end.

Methods of infertility treatment are often mentioned as one of the reasons behind the increased number of twin pregnancies. In the course of the assisted conception process, iatrogenic multiple pregnancies constitute significant and commonly-known complications. The statistical data inform us that 19% of twin pregnancies are connected with in vitro fertilization (IVF), while 21% are connected with methods of assisted conception other than IVF, which together amounts to a considerable 40% [9,10,11,12]. Multiple pregnancy is fraught with a higher risk of problems concerning pregnancy maintenance when compared with that of a singleton. 

Firstly, the overall risk of miscarriage of both fetuses in twin pregnancy is higher than in singleton pregnancy [13,14]. Secondly, the normal course of pregnancy maintenance may be disrupted by preterm delivery [15]. According to a directive of the Polish Ministry of Health, a delivery involving a monoamniotic pregnancy has to be scheduled by the end of the 34th week of pregnancy, for a dichorionic–monoamniotic pregnancy by the end of the 37th week, and for a dichorionic–diamniotic pregnancy by the end of the 39th week. More than half of twin pregnancies lead to preterm delivery [16].

Future parents’ awareness concerning the risks connected with twin pregnancy may significantly affect their sexuality. The stress and fear associated with the possible complications not only negatively affects the quantity and quality of sexual intercourse, but also has a profound impact on the quality of a relationship between partners.

A twin pregnancy implies many potential complications for the fetus itself. For instance, when it comes to cerebral palsy, the risk can be 6.5 times that for singletons [17]. Moreover, the incidence of limited fetus growth occurs ten times more frequently [15]. Other no-less-important complications associated with twin pregnancy are twin-to-twin syndrome (TTTS) and twin reversed arterial perfusion sequence (TRAP), both of which are caused by the presence of vascular anastomoses between fetuses [11].

It should not be forgotten that a twin pregnancy puts a lot of strain on the mother’s body. The stress exerted on the cardiovascular system, higher blood clotting, hormonal changes, and rapid weight gain can lead to many adverse consequences. It also significantly increases the risk of acute complications among future mothers. Indeed, the incidence of severe cases is four times higher if compared to that of a singleton pregnancy [8]. An even greater awareness of complications may be observed among parents who have undergone infertility treatment and in vitro fertilization. Subfertile women tend to develop more complications with diminished gestation time and decreased birth weight, regardless of infertility treatment or adjustments for plurality [9]. Moreover, twins conceived through IVF correlate with the occurrence of bleeding during the first trimester of pregnancy [12].

Due to limited information on the sexuality of women presenting twin pregnancy, we decided to examine this field in more detail. This topic seems to be a taboo even in singleton pregnancy, not to mention multiple pregnancy. Both this, and the fact that it is an area undoubtedly in great need of research, constitutes the reason why we decided to explore this topic at a deeper level.

## 2. Material and Methods

### 2.1. Study Group

Our study sample was derived from the Department of Pregnancy Pathology, the Department of Woman’s Health in the School of Health Sciences at the Medical University of Silesia, Katowice, Poland, between January 2017 and January 2019. In the hospital with the highest level of reference, which is an integral part of the Department of Women’s Health, at the time of the study, the total number of births was 4338, while there were 137 twin deliveries. Pregnant women (*n* = 100) were recruited after a twin pregnancy confirmed by a transvaginal ultrasound and screened for inclusion and exclusion criteria. The gestational age was established from the last menstrual cycle and verified by ultrasound scan measurements. Eligible women who were healthy, pregnant, aged eighteen or older, with no aggravating medical history, and who had given written and informed consent to their participation were included in the study. The exclusion criteria comprised the following: a singleton pregnancy; a multiple pregnancy in which there were more than two fetuses; a history of miscarriage; a suspicion of congenital defects; threatened abortion; placenta previa; cervical incompetence; intrauterine growth restriction; the absence of a sexual partner/lack of sexual activity in the previous four weeks; and, finally, the lack of informed consent of the patient.

Women completed the self-administered questionnaire once in each trimester of pregnancy during consecutive checkups or childbirth classes, namely: (T1) between the 7th and 12th week of gestation; (T2) between the 13th and 24th week of gestation; (T3) between the 25th and 39th week of gestation.

The university’s Ethics Committee waived the requirement for informed consent due to the anonymous and non-interventional nature of the study (KNW/0022/KB/68/19).

The control group (*n* = 624) in our study consisted of the patients from the previous research of Fuchs et al., (2019) about singleton pregnancies. The surveyed women answered exactly the same questions following the same time scheme.

### 2.2. Questionnaire

The questionnaire was split into two parts. The sociodemographic characteristics and gynecological history were collected by a questionnaire. The next part was the female sexual function index (FSFI), the version deemed suitable for Poland. It is a certified survey consisting of 19 questions that evaluates sexual functioning among women. It includes six main subjects, namely desire (questions 1–2, scored 1–5); arousal (questions 3–6, scored 0–5); lubrication (questions 7–10, scored 0–5); orgasm (questions 11–13, scored 0–5); satisfaction (questions 14–16, scored 1–5); and, finally, pain (questions 17–19, scored 0–5) in the period of 4 weeks prior to the survey. The maximum score is 36, and the lowest is 2. A lower score indicates worse sexual functioning.

Additionally, the second part of the survey included concerns about sexual intercourse in different trimesters, as well as its frequency. The survey was tested on a small group, initially consisting of 20 women, in order to evaluate the clarity of the questions.

### 2.3. Statistical Analysis

Analysis of all data was performed with StatSoft Statistica, version 13.0 PL software (Dell, TX, USA), with a rate of *p* < 0.05 being treated as statistically significant. In order to compare the scores between subsequent trimesters, we performed the Friedman ANOVA test as it has been adapted for dependent variables. The post hoc analysis was performed via the Wilcoxon rank test, while to evaluate the quantitative variables, chi-squared was used. Independent variables were calculated with the Kruskal–Wallis test, while the U Mann–Whitney test was used for further analysis. The results are shown as a standard error—mean ± SE.

## 3. Results

Detailed characteristics of the respondents are presented in Table 1. The average age of the surveyed women was 27.6 ± 4.6 years, and 65% of patients were under 30 years old. From the remaining 35% of women over 30 years old, only six women were over 35 years old. Moreover, from all the 100 women subjected to the survey, 58 were married and 35 were in an informal relationship. In addition, seven patients described their matrimonial status as divorced.

Fifty-four percent of the respondents were nulliparous, while 46% of them were in the second or subsequent pregnancy.

In the part of the form consisting of questions concerning education, 40 women (40%) indicated that they had a higher education, 23 (23%) were high school graduates, 17 (17%) had a vocational education, while 20 (20%) women had not finished any school.

According to the answers concerning residence, 35 (35%) respondents lived in a large city with over 250,000 inhabitants, 25 (25%) lived in a city with a population of 50,000 to 250,000, 28 (28%) women lived in towns, while 12 (12%) lived in rural areas.

The mean overall FSFI is presented in Figure 1. A comparison of the average FSFI results for individual trimesters showed a statistically significant deterioration of female sexual function between the first and third trimesters as well as between the second and third trimesters. Statistically significant differences (*p* < 0.05) were reported in both cases. The mean trimester FSFI values were as follows: 25.6 ± 6 in the first trimester; 24.8 ± 7 in the second trimester; and 22.6 ± 8.4 in the third trimester. However, there was no statistically important difference between the first and second trimesters (*p* > 0.05). The mean overall FSFI was estimated to be 24.3 ± 6.1.

As presented in Table 2, no statistically significant difference (*p* > 0.05) was found when comparing mean FSFI obtained in singleton and twin pregnancies. We could observe a similar trend in decrease of FSFI while analyzing respective trimesters in both singleton and twin pregnancies.

Taking into consideration all the domains separately, no significant changes were found when comparing the first and second trimesters of twin pregnancies. A statistically significant decrease in the median value was found for domains orgasm (1 vs. 3, 2 vs. 3 both *p* < 0.05), pain (1 vs. 3 *p* < 0.01, 2 vs. 3 *p* < 0.05), and lubrication (2 vs. 3 *p* < 0.05). For all these domains, the results were the highest in the first trimester and the lowest in the third trimester. The results are presented in Table 3.

There were no differences regarding the age or marital status of the surveyed women. The differences in the FSFI result obtained by women in their first pregnancy and by those in their second or subsequent pregnancy were not statistically significant. The type of multiple pregnancy was also insignificant regarding the FSFI result, with *p* > 0.05 for all of the above.

The results regarding socioeconomic status were not as consistent. Although no statistically significant differences in the FSFI result were found among women with different educational backgrounds (*p* > 0.05), these differences came to the fore in the context of the place of residence, as well as the living and housing conditions of the pregnant women.

The total score of the FSFI differed in the category of place of residence as follows: for the inhabitants of rural areas, it was 21.7 ± 5.4, while for the inhabitants of small and medium towns it was 24.4 ± 3.8 and 25.9 ± 4.9, respectively (in all three *p* < 0.05).

In terms of living conditions, the difference in the value of the mean total FSFI was statistically significant (*p* < 0.001). For the average living conditions, an FSFI of 22.1 ± 5.3 was reported, and for very good living conditions—25.5 ± 5.1. The results are presented in Figure 2.

When we compared all three trimesters of twin pregnancies, no statistically significant changes were found in the mean FSFI score. Moreover, no difference regarding sexual position preferences occurred. In all trimesters of twin pregnancy, women were most likely to choose intercourse in the spoons position, which was confirmed by 38%, 28%, and 45% of women choosing this option in the first, second, and third trimesters, respectively. The other positions often mentioned by the patients were missionary, doggy style, and cowgirl positions. The study results also showed a growing trend in the number of women not having sex at all over the course of a twin pregnancy. In the first trimester, 6% of surveyed women reported no sexual intercourse week-by-week. This percentage grew to 8% in the second trimester and to 15% in the last trimester of pregnancy (Figure 3).

The frequency of sexual intercourses per month depending on the trimester of pregnancy was analyzed as well. However, this parameter was taken into consideration only in sexually active women. The median number of intercourses per month was the highest in the first trimester of pregnancy, rounded up to 13 intercourse per month (12.8) and was significantly higher than in the second and the third trimester (*p* < 0.05, *p* < 0.05). For the second and third trimesters, the median was the same—rounded up to nine intercourses per month (8.5) (*p* > 0.05) (Figure 4).

## 4. Discussion and Conclusions

Taking the most recent definition provided by the World Health Organization into consideration, sexual health is not only the absence of a sexual dysfunction but is also recognized as engendering a general sense of comfort, well-being, safety, and empathy in terms of sexual functioning [1]. It is a vastly complicated process controlled by a subtle balance between hormonal, nervous, and vascular systems. Moreover, one should not forget to take into consideration the patient’s family, religion, socioeconomic status, age, past experiences and—last but not least—their health status [18].

The subject of sexuality in pregnant women is poorly studied. Our department’s former research study from 2019 was the first to raise the topic of sexuality of pregnant women and was conducted on as many as 624 patients [19]. The literature shows us that almost 70% of young parents have not been provided with any professional advice considering their sexual well-being over the course of their pregnancy [3].

To the best of our knowledge, no previous investigation about the influence of twin pregnancy on women’s sexuality has been conducted. We decided to perform this research as a continuation of the above-mentioned study also conducted at the Department of Pregnancy Pathology, the School of Health Sciences in Katowice, at the Medical University of Silesia. Our study involved using the FSFI survey, which is used worldwide and has already achieved strong recognition. Moreover, there is a Polish version of the above-mentioned questionnaire, which met the requirements for validation [20].

### 4.1. First Trimester

The mean overall FSFI during the first trimester was the highest when compared with that of subsequent pregnancy periods, placing itself at 25.6 ± 6. The observed trend is comparable with the previously achieved results during the survey among women with singleton pregnancies which was conducted recently in our department [19]. Although it should be noted that an FSFI below or equal to 26 points indicates sexual dysfunction, it should not be forgotten that during the first trimester of pregnancy, many physiological changes occur [21]. Moreover, a lowered FSFI result often does not necessarily translate into a deterioration of one’s sex life. Therefore, we believe that the results from the first trimester should be considered a reference point when analyzing sexual function while pregnant.

Sexual activity during pregnancy may cause discomfort and a need to find more comfortable sexual positions. Many factors contribute to this, such as urinary bladder sensitivity or breast tenderness. The first trimester of pregnancy is also inevitably connected with nausea and vomiting. These symptoms are directly related to the increase of β-hCG. During multiple pregnancy, the concentration of β-hCG grows even more dynamically, resulting in an even more severe manifestation of the conditions mentioned above [22].

One may assume that the FSFI results may be influenced by the birth rate of a future mother. However, our results did not show any correlation between one’s FSFI and pregnancy history. Similar results were obtained by Anğın et al. (2020) [23]. In their study, they compared the FSFI results achieved by pregnant and nonpregnant women, both nulliparous and multiparous. They proved that parity does not determine the FSFI result. In case of twin pregnancies, this phenomenon is most likely caused by a low incidence of multiple pregnancies occurring subsequently.

For the vast majority of patients, multiple pregnancy is something new, unknown and, from the very beginning, more complicated, regardless of the number of children involved. Consequently, the patients are much more concerned not only about their own health but also about that of the fetuses [11,23,24].

### 4.2. Second Trimester

In Fuchs et al. (2019), a significant rise in sexual activity during the second trimester of pregnancy was documented with corresponding FSFI values [19]. Moreover, it has been clearly shown that there is a strong correlation between increased sexual activity caused by greater sexual interest and, what is more important, a general sense of comfort and security. Another significant finding is that the frequency of intercourse during the second trimester doubled when compared to that of the first one. Such findings are consistent with those obtained by Küçükdurmaz et al. (2016). In our experiment, we achieved completely opposite results, where FSFI during the second trimester was 24.8 ± 7. Although the decrease in FSFI score was not statistically significant when compared with that of the first trimester, we could observe a lowering trend [24]. However, the frequency of intercourse was significantly lower.

Sexual activity among women with a twin pregnancy progressively decreases throughout gestation which may be caused, for instance, by a lower sense of security. General concerns are, of course, understandable, as a twin pregnancy is strongly connected with preterm delivery. In addition, it is considered as a high-risk pregnancy [15]. Therefore, it strongly affects the whole sense of comfort, safety, and well-being, not only of the woman herself, but also of her partner. It should not be forgotten that men also undergo similar psychological distress connected with pregnancy. Although Saotome et al. showed that men’s overall satisfaction in the second trimester of pregnancy is higher when compared with that in the first trimester, it was clearly demonstrated that sexual desire itself was significantly lower in the second trimester [25].

Özgan Çelikel et al. has shown that the reluctance of men towards engaging in sexual intercourse with their spouse in their second trimester of pregnancy can be as high as 18% [26]. This highlights an important topic that sexual problems and sexual distress are not equivalent terms. Having said that, the statistical data show that the majority of women with sexual problems do not report sexual distress [27].

### 4.3. Third Trimester

According to researchers’ views, during the third trimester, the FSFI is unsurprisingly lower when compared with that during the previous trimesters. A twin pregnancy is considered a threatened pregnancy by definition and requires early hospitalization. This factor directly impacts sexual life, especially regarding the opportunity to engage in such activity [28].

The type of twin pregnancy seems to have a particularly significant impact on the results of the FSFI. In a monochorionic–monoamniotic pregnancy, hospitalization is recommended as early as the 26th week of pregnancy. Nevertheless, women with a dichorionic pregnancy should be hospitalized in the third trimester as soon as any signs threatening fetal well-being occur.

A constant fear concerning child safety can lead to reluctance to have sexual intercourse both on the part of the mother and her partner. Moreover, the clinical status of women and early recommended hospitalization can also contribute. Although hospitalization does create a physical barrier, psychological barriers also affect women’s sexuality in the third trimester of a multiple pregnancy.

In the third trimester of a twin pregnancy, a woman may feel unattractive. Moreover, pressure on the bladder and pelvic floor muscles may promote the occurrence of dyspareunia during penetration. In this situation, it is difficult to find a comfortable sexual position for both partners [29].

The surveyed women obtained the lowest FSFI score in the third trimester in the following domains: arousal, lubrication, orgasm, and pain. While the difference in scoring between these domains between the first and second trimesters was slight, the decrease in the third trimester was statistically significant [28].

When we compared our results with the ones from the paper considering a singleton pregnancy, the twin pregnancy did not impact sexual functioning in women more. However, when the third trimester came, 15% of women in our research were not involved in any form of sexual intercourse. The Fuchs et al. study considering singleton pregnancies excluded women who did not have sexual intercourse during the third trimester. In this research, we wanted to keep the whole study group, remembering that twin pregnancy occurs less frequently than a singleton one [19].

When we compared the women in twin pregnancy who continued sexual activity in later trimesters, we could see that the frequency of intercourse was significantly lower when compared to that in a singleton pregnancy. However, no differences in the median number of sexual intercourses was found when comparing the second and third trimesters.

### 4.4. Socioeconomic Factors

No statistically significant FSFI differences were demonstrated while considering the course of pregnancy, nor its duration. However, we did observe that socioeconomic status had a crucial impact on the score.

In our study, the most important factors were one’s place of residence and living conditions. Women who resided in cities scored much higher in the survey, regardless of the population. This was connected with the many facilities which larger cities offer, such as easier access to health care or the possibility to reach a hospital or a doctor’s surgery in a matter of minutes. Good living conditions testify to the good financial situation of the respondents. Thus, patients who can afford private health care visit the doctor much more frequently, sometimes even more often than is advised by health care providers.

Visiting a gynecologist or midwife or attending childbirth classes provides a patient with pregnancy education. The future mother not only becomes more aware of her own health and physiology but also feels safe as she knows that she can rely on specialist care. Moreover, it provides her with the opportunity to have a medical consultation as soon as possible, sometimes even by phone or online.

The literature shows a correlation between the scored FSFI and one’s degree of education. However, in our study, this factor was insignificant. This corresponds perfectly with the study of Özgan Çelikel et al. conducted in 2019. We should remember that possessing a higher level of education does not always translate into higher earnings. What is more, possessing a higher education in, for instance, the fields of the technical sciences or humanities does not necessarily mean that a patient will be more aware or have better knowledge of the physiology and course of a multiple pregnancy. Having said that, this proves that regardless of the degree of one’s education, one has to be educated exactly the same way [26].

### 4.5. Assisted Reproductive Technology

As mentioned above, one may observe the occurrence of more and more iatrogenic multiple pregnancies. Unfortunately, assisted reproductive technology is not covered by public health insurance in Poland. This means that every patient considering stimulation of ovulation, insemination or in vitro fertilization must pay all their expenses. Needless to say, these techniques are very expensive, which is a factor that can also have an impact on the results scored in our study in terms of the living conditions and financial status of the patients.

Those patients who are preparing for in vitro fertilization are well-aware of the possible consequences and complications of the procedure and the course of pregnancy. The incidence of complications in the development of the placenta and uterine bleeding are greatly increased in this group. What is more, the same tendency applies, for instance, to the risk of neonatal death [9]. It has been also observed that a lower birth weight among IVF twins correlates with the incidence of bleeding during the first trimester of pregnancy [12].

Being aware of these complications means that it is easier for one to notice alarming symptoms and to seek advice almost immediately. On the other hand, it shows that possessing a higher financial status does not necessarily translate into a more satisfying sex life over the course of pregnancy.

Stammler-Saffar et al. also stresses the importance of concerns and fears. It should be remembered that assisted reproductive patients are even more concerned about the course of pregnancy and fetal well-being as not only the fertilization had been long-awaited, but they have had to cope with many obstacles to achieve this goal [30].

It seems to be that in future studies considering the sexual life of women with multiple pregnancies, more emphasis should be placed on the method of conception involved. Patients availing of assisted reproductive techniques may considerably lower the obtained results. The introduction of a factor whether the conception was natural or assisted should help achieve much clearer results. This can help select groups of women and their partners who particularly need sex education in terms of sexual intercourse during pregnancy.

The obtained results showed that the topic of sexual intercourse during pregnancy should be discussed in the doctor’s office on the first appointment. Patients often use nonmedical sources, which leads them to propagating and believing in myths, instead of consulting a doctor. They are confused and disorientated and may decide not to have any sexual intercourse during pregnancy at all. This highlights the fact that the early start of sexual education during pregnancy is a very important topic. It may lead to a reduction of stress and an increase in sexual function of both women and their partners.

## Figures and Tables

**Figure 1 ijerph-19-03546-f001:**
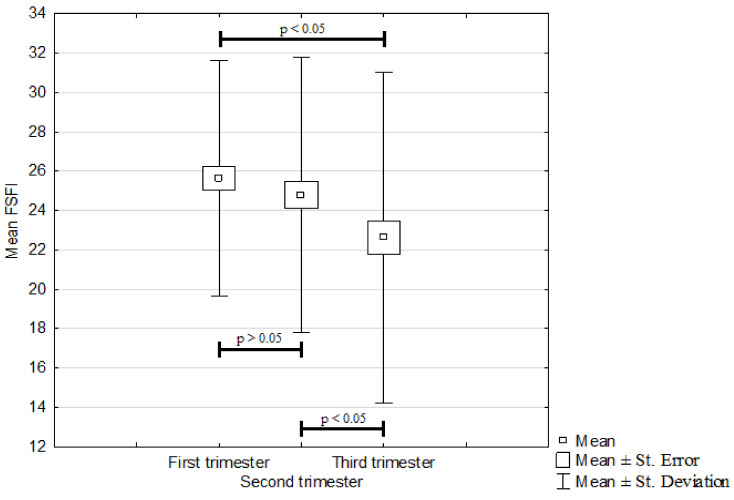
Mean FSFI score depending on pregnancy trimester.

**Figure 2 ijerph-19-03546-f002:**
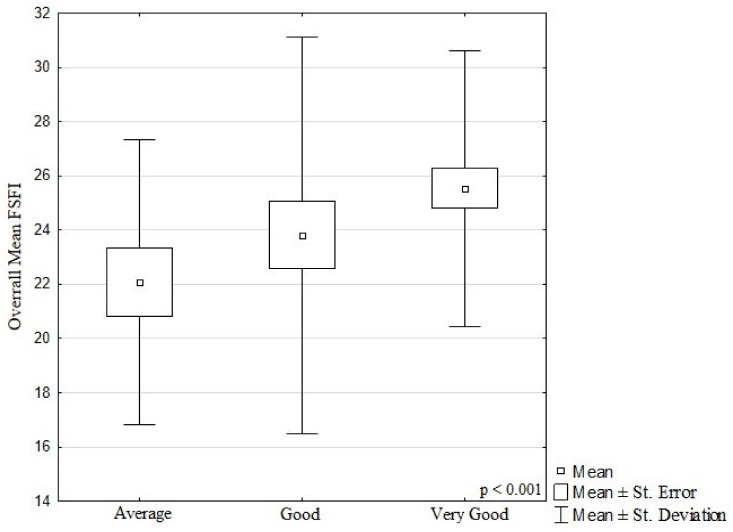
Mean FSFI scores obtained by women in twin pregnancies depending on their living and housing conditions.

**Figure 3 ijerph-19-03546-f003:**
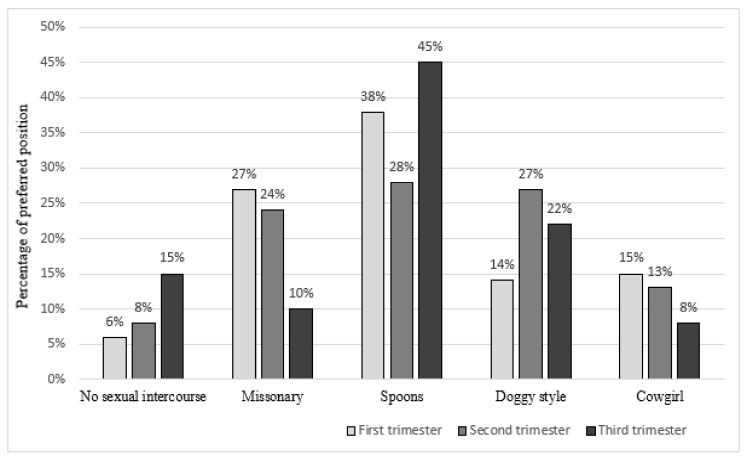
The preference in sex positions depending on twin pregnancy trimester.

**Figure 4 ijerph-19-03546-f004:**
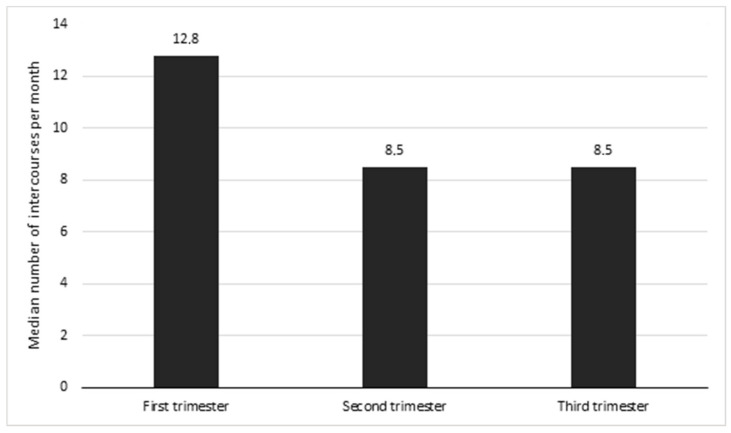
Mean number of sexual intercourses (per month) of women in twin pregnancy.

**Table 1 ijerph-19-03546-t001:** Subjects’ characteristics.

Participant Characteristics	No. (%)
Total respondents	100
**Age (years)**	
Under 30	65 (65)
Over 30	35 (35)
**Marital Status**	
Married	58 (58)
Informal relationship	35 (35)
Divorced	7 (7)
**Education**	
Non-high school graduates	20 (20)
High school graduates	23 (23)
University degree	40 (40)
Vocational	17 (17)
**Place of residence**	
City above 250,000 residents	35 (35)
City with 50,000–250,000 residents	25 (25)
Town below 50,000 residents	28 (28)
Rural settlement/area	12 (12)
**Living and housing conditions**	
Average	17 (17)
Good	35 (35)
Very good	48 (48)
**Parity**	
Primiparity	54 (54)
Multiparity	46 (46)
**Type of pregnancy**	
Monochorionic–Monoamniotic	8 (8)
Monochorionic–Diamniotic	13(13)
Dichorionic–Diamniotic	79 (79)

**Table 2 ijerph-19-03546-t002:** Comparison of mean FSFI scores obtained in singleton and twin pregnancies. The results from single pregnancy are presented in Fuchs et al. (2019).

Variables	Single Pregnancy	Twin Pregnancy	*p*
First trimester	26.1 ± 6.1	25.6 ± 6	>0.05
Second trimester	25.9 ± 8.7	24.8 ± 7	>0.05
Third trimester	22.7 ± 8.7	22.6 ± 8.4	>0.05

**Table 3 ijerph-19-03546-t003:** Comparison of FSFI scores’ medians in the first, second, and third trimester of twin pregnancy.

Variables	First Trimester	Second Trimester	Third Trimester	1 vs. 2	1 vs. 3	2 vs. 3
Desire	4.2 (3.6–4.8)	4.8 (3.6–5.4)	4.2 (3.6–4.8)	>0.05	>0.05	>0.05
Arousal	4.8 (4.2–5.7)	4.8 (3.9–5.7)	4.8 (3.6–5.4)	>0.05	>0.05	>0.05
Lubrication	3.6 (3.6–5.1)	3.9 (3.6–4.7)	3.6 (3.3–4.4)	>0.05	>0.05	<0.05
Orgasm	4.2 (3.6–5.2)	4.4 (3.6–4.8)	4.0 (3.6–4.8)	>0.05	<0.05	<0.05
Satisfaction	4.0 (4.0–5.2)	4.4 (3.6–5.2)	4.2 (3.6–5.2)	>0.05	>0.05	>0.05
Pain	3.6 (2.4–5.0)	3.6 (2.4–4.4)	3.2 (1.6–4.0)	>0.05	<0.01	<0.05
Total FSFI	24.4 (22.3–30.2)	24.8 (21.4–28.5)	24.1(21.1–27.0)	>0.05	<0.05	<0.05

## Data Availability

The original data are available after contact with the corresponding author.

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
