# Peer review of "Female Sexual Function in Twin Pregnancy"

_ijerph, 2022, doi:10.3390/ijerph19063546_

Round 1

Reviewer 1 Report

In this study, Fuchs and coauthors investigate the effects of twin pregnancy on female sexual function throughout the three trimesters of pregnancy. The study is interesting, but some points should be improved prior to publication.

Major points:

  1. Statistical analyses need to be more detailed. For example, tests such as Wilcoxon rank and Mann-Whitney are recommended to compare only two groups while Kruskall-Wallis is recommended to compare more than two groups. Therefore, defining which tests are being used for which comparisons (which parameters) is necessary.
  2. The information of women with singleton pregnancies (used for figure 1) requires more details, including number of women, and should appear in the material and methods.
  3. The text below figure 1 and tables look like text in the results section, but are shown with a margin that makes it look like legends of figure and tables. Please, adjust text to be shown as text of results section instead of legends of figure and tables.
  4. Results described in legend of table 2 (arousal, lubrication, orgasm, and pain) deserve to be shown in table 2.
  5. It is curious that mean values for arousal are the same for each trimester, but the p-value is <.01. Please, clarify. Also, the way the data are presented in table 3 does not allow the reader to visualize what are the differences between each one of the trimesters. This reviewer recommends using the system of letters (for example, 3.6a, 3.9b, 3.6a means the second trimester is different than first and third trimester).
  6. Further clarification is necessary to define what is average, good, and very good housing conditions.
  7. It is not clear whether frequency of sexual intercourses per month (described in lines 199-204) significantly differed. The p-value should be added to text.
  8. The title of figure 4 is “Discussion and conclusions”. This should be corrected.
  9. It is not clear what the authors mean by “low incidence of multiple pregnancies occurring subsequently” (line 253). Is there a reference to support this discussion?
  10. In text in lines 265 to 267, it is stated that FSFI during second trimester is lower than the first trimester, but no significant difference was detected between the first and second trimesters of gestation as shown in figure 1. This requires clarification. Also, if the second trimester did not decrease FSFI significantly, there should be no discussion about possible causes for lower FSFI in the second trimester. Instead of breaking down the discussion for each trimester, this reviewer recommends a different structure for the discussion to address the differences between the third trimester and the other two trimesters.
  11. It seems like no questions about sense of security during pregnancy was asked in the questionnaire. Therefore, although this might be a possibility, the emphasis of the discussion for the differences found in this study should be on socio-economic factors much more than on sense of insecurity or any other factor not tested. This reviewer recommends a deeper discussion about why socioeconomic factors are important in potentially driving the results found in this study.

Minor points:

  1. This reviewer recommends adding the word “twin” prior to pregnancy on line 17 (abstract).
  2. Several parts of the text describe results of previous studies and, thus, should have a reference, including lines 37, 41, 55, 66.
  3. On lines 46-47, it would be beneficial to specify the “phenomenon”. What is this referring to, specifically?
  4. Several paragraphs should be combined with others throughout the manuscript to help in the text flow. For example: the paragraph starting at line 52 should be combined with the next paragraph; the paragraph starting at line 71 should be combined with the next paragraph; the paragraph starting at line 207 should be combined with the next paragraph; the paragraph starting at line 215 should be combined with the next paragraph; the paragraph starting at line 221 should be combined with the next paragraph; the paragraph starting at line 230 should be combined with the next two paragraphs; the paragraph starting at line 240 should be combined with the next paragraph.
  5. The line 187 deserves clarification. What was compared here?

Author Response

Thank you for your time and all your valuable remarks. We allowed ourselves to present the changes and responses to individual comments in points.

Major points:

  1. The Fridman test was used to compare more than two dependent variables, i.e. 1,2,3 trimester. Subsequently, detailed tests for two variables (i.e. between trimesters 1 and 2, 1 and 3 and 2 and 3) were used the Wilkoxon test. In the case of more than two independent variables, the Kruskalwallis test was used. The analysis was then extended to include the Mann-Whitney test between the two variables. Information on the tests used has been supplemented in the materials and methods section.
  2. This is a continuation of the research, therefore we would not like to publish specified data, as if it has already been published before. We followed your advice and added basic information about the group in the material and methods.
  3. Thank you for your suggestions for point 3. Necessary changes were provided.
  4. For point 4 necessary changes were provided.
  5. For point 5 necessary changes were provided.
  6. It is the responses to questions about housing conditions were not deliberately defined. The answers are to be the subjective opinion of the respondents.
  7. We have made appropriate changes to Table 3 as well as the p value was added to the text.
  8. For point 8 necessary changes were provided.
  9. For point 9 necessary changes were provided.
  10. Thank you for your excellent point. We haven’t found the papers about the occurrence of subsequent multiple pregnancies per se, but we provided the information about the low rate of twin deliveries. Therefore, the chance of subsequent multiple pregnancies seems to be low.
  11. Although we have not achieved statistically significant results, we can observe a trend. In the different papers a rise in FSFI is observed in the second trimester of pregnancy. In our study, regarding multiple pregnancies, a lack of typical FSFI increase should be considered as a negative effect. Therefore, we discussed the possible causes for such FSFI. Your remark allowed us to improve this part.
  12. Thank you, we emphasised the importance of socioeconomic factors, including money and the availability of medical service.  We also emphasised that ART is very expensive. 

Minor points:

Thank you, we have followed all your suggestions regarding minor points (1-5) and improved the text.

Reviewer 2 Report

The manuscript ”Female sexual function in twin pregnancy” is an elegant study that looked at sexual function patterns in women with twin pregnancy. The study compares population level satisfaction in women at different stages of gestation. A nice feature of the manuscript is that it seems to have been written following the recommendations by STROBE, so it is easy to assess what was done.

Given the detailed descriptions in the text, it probably would be best to add table 1 as a supplementary online file. 

Similarly, Table 2 and Figure 1 are repetitive, so just keep one in the main manuscript text.

The categories of Figure 2 (in the x axis) need to be consicely defined in the methods since it is not clear to what do they refer.

Figure 4 misses a measure of the variability in the number of intercourses. Please add a mark indicating the SD in the response.

Overall the manuscript requires a moderate amount of english editing, for clarity and brevity.

Author Response

Thank you for your time and all your valuable remarks. We allowed ourselves to present the changes and responses to individual comments in points.

  1. We think that the data is clearer and more visible if shown in the table. Although if the editor would prefer the supplementary online file form, we will follow his advice.
  2. Figure 1 shows the comparison between the trimesters of twin pregnancy, while table 2 shows the comparison between relevant trimesters in singleton vs. twin pregnancy.
  3. Thank you, we improved the text following your advice.
  4. Thank you, necessary changes were provided and the p value is now described in the text. 

Reviewer 3 Report

This is an interesting descriptive study that highlights a significant gap in the public health area. However, it needs major revisions before it can be considered for publication.

In some parts the writing style does not sound appropriate for publishing in a scientific journal and is closer to a newspaper article style of writing. Therefore, I suggest receiving help for academic writing and editing.

The study design seems to be a prospective cohort as researchers followed the women over their entire pregnancy and collected data at different time points. However, authors refer to it as a “cross-sectional” study. Please consider revision or further elaboration on how it is a cross-sectional study.

Pregnant women (n=100) were recruited personally after a twin pregnancy was confirmed by transvaginal ultrasound and screened for inclusion and exclusion criteria.

What does “personally” mean here?

The questionnaire was split into two parts. Firstly, it consisted of questions concerning demographic and social aspects, as well as the patient’s gynaecological history.

Concerning? Please revise the sentence to something like “The sociodemographic characteristics and gynaecological history were collected by a questionnaire.”

Analyses with a small sample size (100) and comparison of categories can reduce the precision of the results. Understandably, the prevalence of twin pregnancy is low, but the authors may have been able to collect more data during the past two years spent improving the manuscript. Still, with the same sample size, strengthening the results with appropriate statistical analyses could help. The statistical approach used in this study cannot determine the impact of any independent variables such as twin pregnancy or living in rural areas. To measure the impact or association, regression models are needed. Examples of statements on the influence of factors are line 14: “The patient’s place of residence had a crucial impact on their FSFI score.” and line 221: To the best of our knowledge, no previous investigation on the subject of the influence of twin pregnancy on women’s sexuality has been conducted.” The current study cannot draw conclusions about association or impact of these factors, it can only highlight differences.

In addition, the outcome of interest -female sexual function- can be impacted by confounders. The authors did not choose a statistical approach that allows adjustment for potential confounding factors. Therefore, unassessed factors could influence the results. This can be solved by using regression analysis adjusted for confounding factors, otherwise it should be mentioned as a limitation for the study.

Another issue with the analyses is that the authors used Friedman and Wilcoxon rank tests for analysing outcomes. The use of these tests shows that the data were not normally distributed. Although for instance, Friedman test compares the mean ranks between groups, authors need to report the median value for each group. Therefore, tables 2 and 3 need revision.

Table 2 included singleton and twin pregnancies, but it is unclear why there is no comparison in table 3 if the study is comparing twin pregnancy with singleton pregnancy.

Title of figure 4 needs revision. Please select an appropriate title relevant to the title of the Y axis.

Some statements are indirect and need academic editing. An example is: line 300-301: Possessing a big belly, movement difficulties and swelling do not have a positive effect on a pregnant woman's libido. This sentence could be shorter and clearer like “A big belly, movement difficulties, and swelling negatively impact pregnant women's libido (cite the appropriate reference/s if there’s any).”

If we compare our results with the ones from the paper considering the singleton pregnancy, the twin pregnancy does not impact sexual functioning in women more. However, when the third trimester came 15% of women in our research did not involve in any form of sexual intercourse.

Where are the results from the paper considering the singleton pregnancy? Do authors expect the reader to stop reading their current article at this stage and read the entire other publication? Although it is expected that readers check the relevant cited references later, an applicable practice is discussing the previous data further here.

The Fuchs et al. study considering singleton pregnancies did not include women who did not have any sexual intercourse during the third trimester. In this research we wanted to keep the study group as numerous as possible, remembering that twin pregnancy occurs less frequently than singleton one. Reduction of patients included in the research could indeed affect the results, therefore we did not exclude patients who did not have any sexual intercourse in all the trimesters [14].

Please revise the above lines. For example, what does “keep the study group as numerous as possible” mean?

The discussion section is rich with information and the authors have pointed out a significant gap in knowledge. However, the writing style is not entirely scientific. I strongly recommend that authors receive help to improve their writing. An example is: line 377-383 “On the basis of the achieved results, we believe that the topic of sexual intercourse during pregnancy should be discussed in the doctor’s office as soon as possible. Through embarrassment, patients often seek help from non-medical sources which results in them propagating worthless myths. Indeed, they are often so confused and disorientated that they may decide not to have any sexual intercourse during pregnancy at all. This highlights the fact that an early start in sexual education during pregnancy may lead to a reduction in stress and an increase in sexual function for both women with a twin pregnancy and their partners.”

Author Response

Thank you for your time and all your valuable remarks. We allowed ourselves to present the changes and responses to individual comments in points.

  1. Thank you, we improved the text following your advice
  2. We agree with your opinion and thank you very much for your attention. Unfortunately, we missed this error in the creative process. It is a prospective, not cross-sectional, cohort study
  3. Thank you, we improved the text following your advice
  4. For point 4 necessary changes were provided.
  5. Multilayer analyzes were not used in the study. Only the majority of factors that could modify the result of sexual activity in twin pregnant women were excluded, eg education, material conditions, etc. (p> 0.05). We agree with your view that the methodology really only allows us to spot the problem and highlight the differences.
  6. 6. Unfortunately, the study did not use confounder-adjusted regression analysis. We are awere of the fact that for this reason, this topic requires further research.
  7. We have made appropriate changes to the tables.
  8. Originally, the work was not supposed to compare single and twin pregnancies, but only present the problem of twin pregnancies. In the previous review process, however, such comparisons were requested from us. The results have been revised and approved by previous reviewers.
  9. For point 9 necessary changes were provided.
  10. Thank you, we improved the text following your advice
  11. This is a continuation of the research, therefore we would not like to publish specified data, as if it has already been published before. We followed your advice and added basic information about the group in the material and methods and provided additional information in discussion. No further knowledge is required to compare singleton vs. twin pregnancy.
  12. Thank you, we reviewed the English and clarified this part.
  13. Thank you, we reviewed the English and the writing style.

This manuscript is a resubmission of an earlier submission. The following is a list of the peer review reports and author responses from that submission.